# Generic approach for mathematical model of multi-strain pandemics

**Teddy Lazebnik** [1]*, **Svetlana Bunimovich-Mendrazitsky** [2]

**1** Department of Cancer Biology, Cancer Institute, University College London, United Kingdom, **2** Department of Mathematics, Ariel University, Ariel, Israel

* t.lazebnik@ucl.ac.uk

**Data Availability Statement:** All files are available from the WHO's COVID database at https://covid19.who.int/.

## Abstract

Multi-strain pandemics have emerged as a major concern. We introduce a new model for assessing the connection between multi-strain pandemics and mortality rate, basic reproduction number, and maximum infected individuals. The proposed model provides a general mathematical approach for representing multi-strain pandemics, generalizing for an arbitrary number of strains. We show that the proposed model fits well with epidemiological historical world health data over a long time period. From a theoretical point of view, we show that the increasing number of strains increases logarithmically the maximum number of infected individuals and the mean mortality rate. Moreover, the mean basic reproduction number is statistically identical to the single, most aggressive pandemic strain for multi-strain pandemics.

## 1 Introduction and related work

Over the centuries, humanity has experienced multiple types of disasters [1–4]. One of them is pandemics (local and global) that cause significant mortality [5]. Moreover, recent studies show that the occurrence rate of new pandemics has increased in the last century, resulting in an increased number of pandemics and their influence [6]. Some of these pandemics exert a global influence such as HIV/AIDS that killed 680 thousand individuals only in 2020 according to the World Health Organization (WHO) or the COVID-19 pandemic that killed 4.5 million individuals and infected around 440 million individuals worldwide during its first 18-months starting in early 2019 [7]. As a result, the need for policymakers to be able to control the spread of a pandemic is becoming more relevant by the day [8].

Moreover, due to multiple socioeconomic processes, there is an increase in the speed at which new infections are spread [9]. To be exact, globalization has facilitated strain spread among countries through the growth of trade and travel [10]. Diseases are usually caused by pathogenic agents, including viruses and bacteria, which can be denoted as multiple variants, generally named strains. The emergence of a multi-strain pathogen imposes a new challenge to control the spread of disease [11]. Since new strains occur as it reproduces in new hosts, the large population of infected individuals offers a fertile ground for new strains to appear [12,

**Funding:** The authors received no specific funding for this work.

**Competing interests:** The authors have declared that no competing interests exist.

13]. For example, in the case of COVID-19, already in the first year and a half of the pandemic, four (globally common) strains were detected [7].

Most diseases have several pathogenic strains, which can make it difficult to fight the disease and lead to complex dynamics. However, their dynamic properties have not been adequately studied [11]. Hence, a better understanding of future pandemics with several strains is a necessary step to ensure the ability of the global community in handling the next pandemic. One approach to tackle this challenge is using epidemiological-mathematical models, which allows us to simulate and investigate multiple scenarios in a safe, cheap, and manageable environment. A large portion of these epidemiological models are based on the Susceptible-Infectious-Removed (*SIR*) model [14]. Over the years, researchers have introduced different extensions to the *SIR* model in order to obtain a more accurate model for biological [15], economic [16, 17] spatial [18–20], and pandemic management [8, 21, 22] properties of a particular disease or socio-epidemiological scenario. These extensions are natural as the *SIR* disease transmission model is derived assuming multiple strong assumptions. For example, the *SIR* model assumes that the population is large and dense or that the infection rate is constant [14]. The authors extend this basic model in many directions by relaxing some assumptions. As such, the mathematical analysis quickly becomes significantly more sophisticated [23].

Cooper et al. [24] used the *SIR* model on the COVID-19 pandemic while relaxing the assumption that the population is mixing homogeneously and that the total population is constant in time. The authors show that the model has a fair fitting on six countries (China, South Korea, India, Australia, USA, Italy).

Another extension of the *SIR* model for the Polio pandemic is proposed by Agarwal and Bhadauria [25]. The authors introduced the fourth stage—vaccinated individuals, resulting in a *SIRV* model. The numerical simulation of the model results in a promising outcome. Nonetheless, the evaluation is limited to a small size (up to a few hundred individuals), and the generalization to larger populations can be less accurate due to the increased chance that a strain occurs during the pandemic and changes its dynamics [13].

Similarly, Bunimovich-Mendrazitsky and Stone [26] proposed a two-age group, extension (adults and children), for the Polio pandemic spread. Using the model in [26], the extraordinary jump in the number of paralytic polio cases that emerged at the beginning of the 20th century can be explained. The model does not take into consideration some strains of Polio [27] which results in an increased divergence from the actual dynamics over time.

In addition, one of the main extensions of the *SIR* model is the *SIRD* (D-Dead) model, as this model is able to represent the reinfection process and the death of individuals due to the pandemic [28–30]. This model better represents the biological-clinical dynamics in human populations as the long-term immunity memory is reduced over time making the individual susceptible again [31, 32]. We based our model on this extension as it allows reinfection in several strains of the original strain.

The mentioned models and other models that extend the *SIR* model can fairly fit and predict the course of a pandemic's spread [33, 34]. However, the models are not fitted to capture sharp changes in the dynamics due to pandemic modifications. One reason is the lack of modelization in multi-strain pandemics.

Indeed, the occurrence of pandemics with multiple mutations is common. For example, Minayev and Ferguson [35] investigate the interaction between epidemiological and evolutionary dynamics for antigenically variable pathogens. The authors proposed a set of relatively simple deterministic models of the transmission dynamics of multi-strain pathogens which provide increased biological realism. However, these models assume clinical-epidemiological dynamics that hold only for a subset of pathogens with cross-immunity of less than 0.4 [35]. In a similar manner, Dang et al. [36] developed a multi-scale immuno-epidemiological model of

influenza viruses including direct and environmental transmission. The authors showed how two time-since-infection structural variables outperform classical SIR models of influenza. During the modelization, they used a within-host model that holds only for the influenza pandemic. In addition, Gordo et al. [12] proposed a *SIRS* model with reinfection and selection with two strains. The authors used a metapopulation of individuals where each individual is depicted as a vector in the metapopulation. This model has been validated on the influenza pandemic in the State of New York (USA), based on the genetic diversity of influenza gathered between 1993 and 2006, showing superior results compared to other *SIR*-based models [12]. Nonetheless, the sophistication of the model is both in its strength and shortcoming, from an analytical point of view, due to its stochastic and chaotic nature.

Moreover, the usage of multi-strain models that are used for specific pathogens is not restricted to influenza. Marquioni and de Aguiar [37] proposed a model where a pandemic starts with a single strain and the other strains occur in a stochastic manner as a by-product of the infection. The authors fitted their model onto the COVID-19 pandemic in China showing improved results when strain dynamics are taken into consideration compared to the other case [37]. Likewise, Khayar and Allali [38] proposed a *SEIR* (E-exposed) model for the COVID-19 pandemic with two strains. The authors analyzed the influence of the delay between exposure and becoming infectious on several epidemiological properties. Furthermore, they proposed an extension to the model (in the *Single and two mutations model* S1 Appendix) for multi-strain dynamics. In their model, an individual can be infected only once and develop immunity to all strains [38]. In our model, we relax this assumption, allowing individuals to be infected once by each strain. Comparably, Gubar et al. [39] proposed an extended *SIR* model with two strains with different infection and recovery rates. The authors considered a group of latent individuals who are already infected but do not have any clinical symptoms.

In addition, Arruda et al. [40] proposed an *SEIR* model with an arbitrary number of mutations and reinfection of the same strain dynamics. The authors proposed an optimal control for the non-pharmacological lockdown policy and validated their model (with and without mitigation) on the COVID-19 pandemic for both England and the state of Amazonas, Brazil. The authors showed that their model can derive optimal mitigation strategies for any number of viral strains, whilst also evaluating the effect of distinct mitigation costs on the infection levels. On the one hand, Arruda et al.'s model takes into consideration an exposed phase (which is commonly found in multiple pandemics [38, 41]) and reinfection of the same strain after some period of time which are not included in our model. On the other hand, their model does not take into consideration the order of infection by different strains which is one of the main contributions of our model.

Furthermore, Fudolig et al. [42] proposed a multi-strain *SIR* based model with selective immunity by vaccination. The authors examined the influence of the introduction of a new strain. In particular, the authors examined the case where a new strain emerges in the population while the preexisting strain is near to extinction or reached a global equilibrium. The emergence of strains during the pandemic rather at the beginning, as suggested by the proposed model, is more realistic. However, it is not in the scope of the proposed model which aims to study the properties of a static number of strains.

Correspondingly, Aleta et al. [43] extended the SIRS model on a metapopulation where individuals are distributed in sub-populations connected via a network of mobility flows. They show that spatial fragmentation and mobility play a key role in the persistence of the disease the maximum of which is reached at intermediate mobility values. Their model assumes a fixed number of locations (using a graph-based model) such that each location has a unique strain-like simulation. Furthermore, Di Giamberardino et al. [44] proposed a multi-group

model formed by interconnected SEIR-like structures which include asymptomatic infected individuals. The authors fitted the data to the COVID-19 pandemic in Italy to study the influence of different IPs on the pandemic spread. The interconnection between the groups in the model is represented by the mobility of individuals between them. The model somewhat represents multi-strains as each group has different epidemiological parameter values and the transformation between them. However, the authors do not handle the case where an individual has been infected by one strain and later infected by others which are known from multiple clinical and biological studies [45–48]. Khyar and Allali [38] studied the global stability of the two-strain epidemic model, extending the *SEIR* with two types of exposed and infected individuals, with two general incidence functions. The authors investigate the basic reproduction number of each strain separately and its effect on the disease-free equilibrium. Roche et al. [49] proposed a stochastic individual-based model for avian influenza viruses, implemented using the agent-based approach. The authors show that their model extends the stochastic SIR model for multi-strain pandemics. Nevertheless, this approach is stochastic in nature, which makes the analytical investigation difficult for multiple pandemic parameters, such as stability and bifurcation.

In this research, we developed an extension of the *SIRD*-based model which allows an arbitrary number of strains $|M|$ that originated from a single strain and is generic for any type of pathogen. The model allows each strain to have its unique epidemiological properties. In addition, we developed a computer simulation that provides an *in silico* tool for evaluating several epidemiological properties such as the mortality rate, max infections, and average basic reproduction number of a pandemic. The proposed model allows for a more accurate investigation of the epidemiological dynamics while keeping the data required to use the model relatively low. The main contribution of the proposed model compared to other *SIR*-based multi-strain models is two-fold: the proposed model does not assume any pathogen-specific properties keeping it as generic as possible by the standard *SIR* model and the order of infection from different strains is taken into consideration.

This paper is organized as follows: In Section 2, we introduce our multi-strain epidemiological model. Based on the model, we present a numerical analysis of three epidemiological properties as a function of the number of strains ($|M|$). In Section 3, we present the implementation of the model for the case of two strains ($|M| = 2$) and provide an analytical analysis of the stable equilibria states of the model and a basic reproduction number analysis. Afterward, we show the ability of the model to fit historical epidemiological data known to have two strains. In Section 4, we discuss the main advantages and limitations of the model and propose future work.

## 2 Multi-strain model

The multi-strain epidemiological model considers a constant population with a fixed number of individuals *N*. We assume a pandemic has $M := \{1, \ldots, m\}$ strains. Moreover, two options are possible: a) strains $[2, \ldots m]$ are mutations arising from one pathogen as a result of the mutation process; b) the disease is characterized by the emergence of *m* pathogenic strains but an individual cannot be infected by more than one strain of the virus at a time.

Each individual belongs to one of the three groups: 1) Infectious with strain $i \in M$ and history of recoveries $J \in P(M)$ (the power set of the strain and its strain set) represented by $R_J I_i$, which maps to the infection (I) state in the *SIRD* model; 2) Recovered with history $J \in P(M)$ represented by $R_J$, which maps to the recovered (R) state in the *SIRD* model; and 3) Dead (*D*) such that

$$N = \Sigma_{J \in P(M \setminus \{i\}), i \in M}(R_J I_i(t)) + \Sigma_{J \in P(M)}(R_J(t)) + D(t), \tag{1}$$

where $i \in M$ is the index of a strain and $J \in P(M)$ is the set of strains an individual already suffered from. For example, $R_\emptyset$ is the group of individuals that do not have a recovery history and are susceptible to all $|M|$ strains which is a private case of $R_J$ where $J = \emptyset$, which is isomorphic to the susceptible (S) state in the *SIRD* model. The proposed model for $|M| = 1$ is isomorphic to the *SIRD* model (the proof is provided in the Section 2 in S1 Appendix). A schematic transition between disease stages of an individual is shown in Fig 1.

Individuals in the Recovered ($R_J$) group have immunity for the strains $k \in J$ and are susceptible to the infection strains $M \backslash J$. When an individual in this group is exposed to a strain $i \in M \backslash J$, the individual is transferred to the Infectious with history of recoveries group ($R_J I_i$) at a rate $\beta_{J,i}$. The individual stays in this group on average $\gamma_{J,i}$ days, after which the individual is transferred to the Recovered group ($R_{J \cup \{i\}}$) or the Dead group ($D$). Therefore, at a rate of $(1 - \phi_{J \backslash i})$, of infection by strain $i$ with a history of recoveries from strains $J$, individuals remain seriously ill or die while others recover. The recovered are again healthy, no longer contagious, and immune from future infection of the same strain. The epidemiological dynamics are described in Eqs (2)–(4).

In Eq (2), $\frac{dR_J I_i(t)}{dt}$ is the dynamical amount of individuals that recovered from a group of strains $J$ and are infected with a strain $i$ over time. It is affected by the following two terms. First, individuals who recovered from group $J$ of strains become infected with strain $i$, with rate $\beta_{J,i}$. These individuals can be infected by any individual with a strain $i$ who has recovered from any group $K$ of strains so that $i \notin K$. Second, individuals recover from strains $J \cup \{i\}$ with rate $\gamma_{J,i}$. For each strain $i$, the group $i$ can be any subgroup of the group $M$, so that $i \notin J$.

$$\frac{dR_J I_i(t)}{dt} = -\gamma_{J,i} R_J I_i(t) + \beta_{J,i} R_J(t) \sum_{K \in P(M), i \notin K} R_K I_i(t). \tag{2}$$

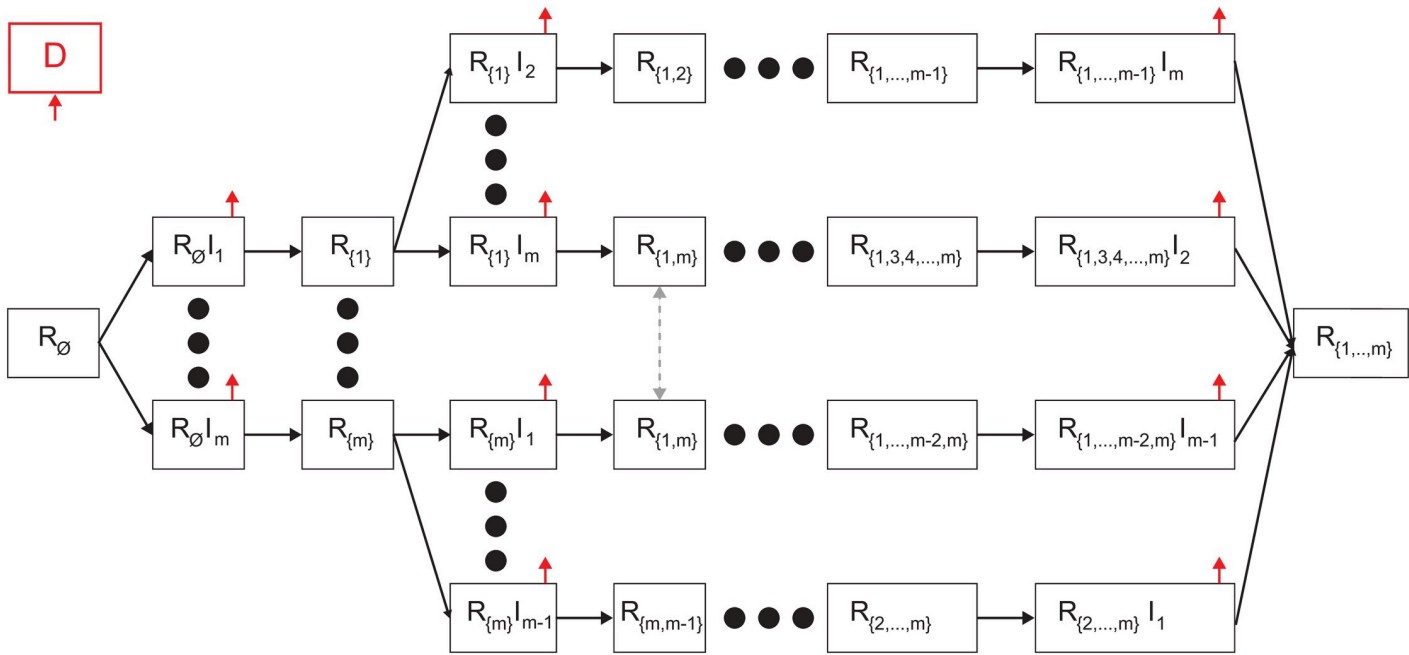

**Fig 1. Schematic view of transition between disease stages.** The red arrows indicates that individuals from the source stage can be transferred to the dead stage. Individuals in $R_J I_i$ stages are necessarily transferred to the respective $R_{J \cup i}$ stages (or dead stage), while individuals in the $R_J$ stages move to $R_J I_l$ stage if they are infected by an individual that is infectious in strain $l \in M$.

In Eq (3), $\frac{dR_J(t)}{dt}$ is the dynamical amount of individuals that recovered from a group of strains $J \in P(M)$ over time. It is affected by the following two terms. First, for each strain $i \in J$, an individual who has recovered from group $J \setminus \{i\}$ of strains and is infected with strain $i$, recovers at rate $\gamma_{J \setminus \{i\}, i}$ with probability of $\phi_{J \setminus \{i\}, i}$. Second, individuals infected by strain $i$ with rate $\beta_{J,i}$. These individuals can be infected by any individual with a strain $i$ who has recovered from any group $K$ of strains, so that $i \notin K$.

$$\frac{dR_J(t)}{dt} = \sum_{i \in J} \left( \gamma_{J \setminus \{i\}, i} \phi_{J \setminus \{i\}, i} R_{J \setminus \{i\}} I_i(t) \right) - \sum_{i \in M \setminus J} \left( \beta_{J,i} R_J(t) \sum_{K \in P(M), i \notin K} R_K I_i(t) \right). \tag{3}$$

In Eq (4), $\frac{dD(t)}{dt}$ is the dynamical amount of dead individuals over time. For each strain $i$, and for each group $J \setminus \{i\}$, infected individuals that do not recover are dying at rate $\gamma_{J \setminus \{i\}, i}$ with the complete probability $(1 - \phi_{J \setminus \{i\}, i})$.

$$\frac{dD(t)}{dt} = \sum_{i \in M, J \in P(M)} \gamma_{J \setminus \{i\}, i} (1 - \phi_{J \setminus \{i\}, i}) R_{J \setminus \{i\}} I_i(t). \tag{4}$$

The dynamics of Eqs (2)–(4) are summarized in Eq (5).

$$\begin{aligned}
\frac{dR_J I_i(t)}{dt} &= -\gamma_{J,i} R_J I_i(t) + \beta_{J,i} R_J(t) \sum_{K \in P(M), i \in K} R_K I_i(t), \\
\frac{dR_J(t)}{dt} &= \sum_{i \in J} (\gamma_{J \setminus \{i\}, i} \phi_{J \setminus \{i\}, i} R_{J \setminus \{i\}} I_i(t)) - \sum_{i \in M \setminus J} (\beta_{J,i} R_J(t) \sum_{K \in P(M), i \in K} R_K I_i(t)), \\
\frac{dD(t)}{dt} &= \sum_{i \in M, J \in P(M)} \gamma_{J \setminus \{i\}, i} (1 - \phi_{J \setminus \{i\}, i}) R_{J \setminus \{i\}} I_i(t),
\end{aligned} \tag{5}$$

The initial conditions of Eq (5) are defined for the beginning of a pandemic as follows:

$$R_{\emptyset}(0) = N - m, \ \forall i \in M : R_{\emptyset} I_i(0) = 1, \ \forall J \in P(M) \setminus \emptyset \wedge i \in M \setminus J : R_J(0) = R_J I_i(0) = 0, \ D(0) = 0. \tag{6}$$

## 2.1 Epidemiological properties

Based on the proposed model, and since for all the cases where $|M| > 2$ it is extremely hard (or even impossible) to obtain an analytical result, we evaluated three important epidemiological properties to see the influence of the number of strains on the pandemic spread: mean basic reproduction number [50], mortality rate [51, 52], and a maximum number of the infectious [16, 53]. Formally, these properties can be defined as follows.

First, the *mean basic reproduction number* is the mean of the basic reproduction number over time during the course of the pandemic. Therefore, it takes the form:

$$E[R_0(t)] := E\left[ \forall J \in P(M) : \Sigma_{i \in M} \left( \frac{R_J I_i(t+1) - R_J I_i(t)}{R_{J,i}(t+1) - R_{J,i}(t)} \right) \right].$$

Second, the *mortality rate* is defined as the number of deaths due to the pandemic divided by the number of infections at some period of time. If not stated otherwise, we assume the mortality rate refers to the entire duration of the pandemic. Hence, it takes the form:

$$\text{mortality rate}(t_0, t_1) := \frac{D(t_1) - D(t_0)}{\Sigma_{J \in P(M)} |J| * (R_J(t_1) - R_J(t_0))}.$$

Finally, the *maximum number of infectious* refers to the cumulative number of infections that

occur during the pandemic. Thus, it takes the form:

$$\text{maximum number of infectious}(t_0, t_1) := \max_{t \in [t_0, t_1]} (\forall J \in P(M) : \Sigma_{i \in M}(R_J I_i(t))).$$

In addition, we define the *most aggressive* strain using the following metric: a strain $k$ is considered more aggressive than strain $l$ if and only if:

$$||[\forall J \in P(M) : (\beta_{J,k}, 1 - \gamma_{J,k}, 1 - \phi_{J,k})]|| > ||[\forall J \in P(M) : (\beta_{J,l}, 1 - \gamma_{J,l}, 1 - \phi_{J,l})]||,$$

using the $L_3$ norm. The motivation of this metric is that a higher infection rate, longer recovery rate, and higher death rate are associated with a more aggressive strain. However, due to the complexity of the pandemic's spread dynamics, it is not straightforward which one of these properties is more important if any, and therefore the comparison between two strains is performed on the three properties simultaneously.

## 2.2 Numerical simulation

Using numerical simulation we aim to study the connection between the number of strains $|M|$ and the proposed epidemiological properties. We numerically solved the model presented in Eq (5) for the case where $|M| \in [1, \ldots, 10]$ using the fourth-order Runge-Kutta algorithm [54]. The model parameters are generated randomly as follows. The infection rates $\beta_{J,i}$ are uniformly distributed in [0.01, 0.10]; the recovery rates $\gamma_{J,i}$ are uniformly distributed in [0.03, 0.33]; and the recovery probabilities $\phi_{J,i}$ are uniformly distributed in [0.90, 0.99] for each strain. The ranges were picked to simulate a large space of possible pandemics, without taking into consideration biological properties associated with cross-immunity between strains. In addition, we assume the population size is 10 million individuals to approximate (in order of magnitude) a European metropolitan area. The simulation begins with one person getting infected by each strain. In addition, it is assumed that no individuals have recovered or died at the beginning of the pandemic. Formally, the initial conditions take the form:

$$R_\emptyset(0) = N - |M|, \ \forall i \in M : R_\emptyset I_i(0) = 1, \ D(0) = 0, \ \forall J \in P(M) \backslash \emptyset, i \in M : R_J I_i = R_J = 0.$$

Moreover, due to the stochastic nature of the simulation originating in the large ranges of values allocated to the model parameters, the simulation is repeated 1000 times, and the mean ± standard division is presented. Using this generation we compute the connection between $|M|$ and the mean basic reproduction number, max infected individuals, and mean mortality rate.

The *mean basic reproduction number* ($E[R_0]$) has been evaluated for each simulation, divided into two cases: the case where each strain has unique parameter values and the case where the parameters of these strains are replaced with the parameter value of the most aggressive strain, defined in Section 2.1, as shown in Fig 2.

The *maximum number of infected individuals* as a function of the number of strains ($|M|$) has been computed and shown in Fig 3. The solid (black) line with the dots represents the numerically calculated values with one standard deviation. Moreover, the fitting function is calculated using the least mean square (LMS) method [55] and shown as the dashed (blue) line. In order to use the LMS method, one needs to define the family function approximating a function. The family function that has been chosen is $f(m) = p_1 \log(m) + p_2$, resulting in

$$E[R_0](m) = 0.103 \log(m) + 0.068. \tag{7}$$

The fitting function was obtained with a coefficient of determination $R^2 = 0.79$.

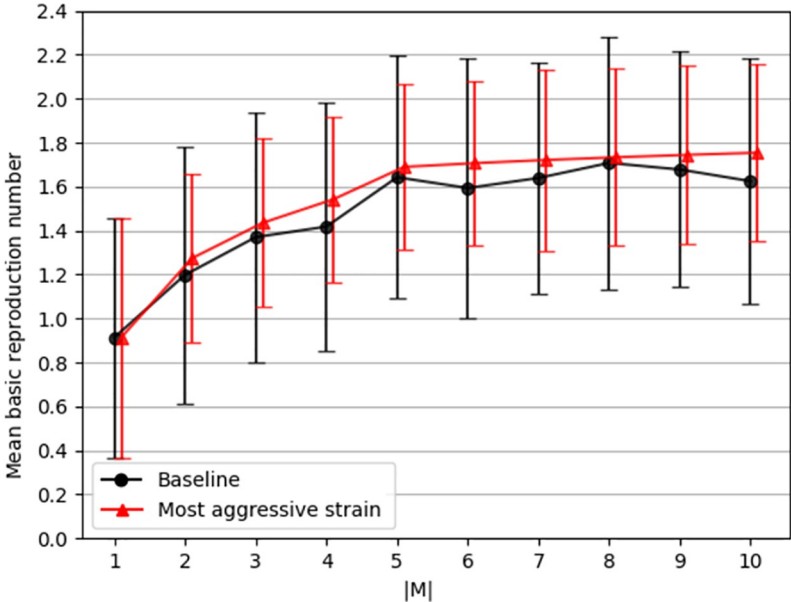

**Fig 2. The mean base reproduction number ($E[R_0(t)]$) as a function of the number of strains ($|M|$).** The black (with circle markers) line indicates the baseline dynamics of the simulation where each strain has unique parameter values. On the other hand, the red (with triangle markers) line indicates the case where all the strain parameters values have been replaced with one of the most aggressive strains. The results are mean ± standard division for $n = 1000$ repetitions.

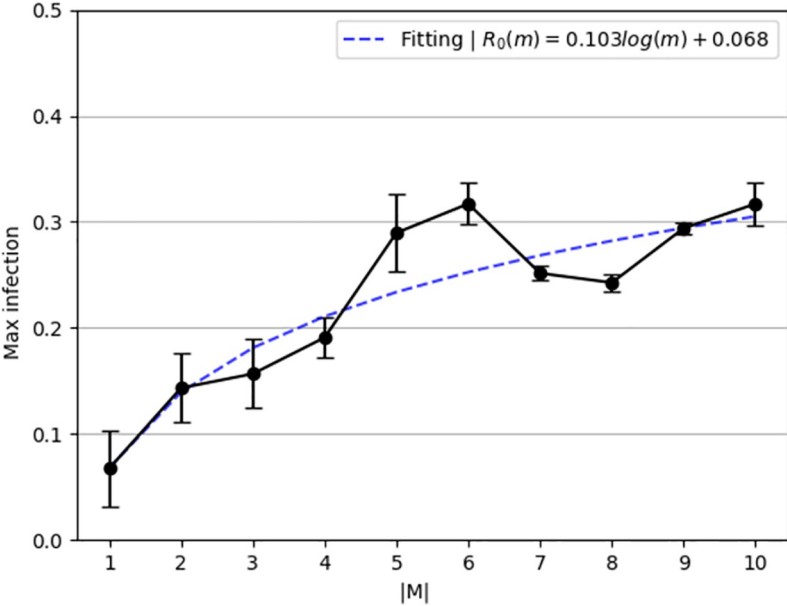

**Fig 3. Maximum number of infectious individuals at the same time as a function of the number of strains ($|M|$).** The results are mean ± standard division for $n = 1000$ repetitions.

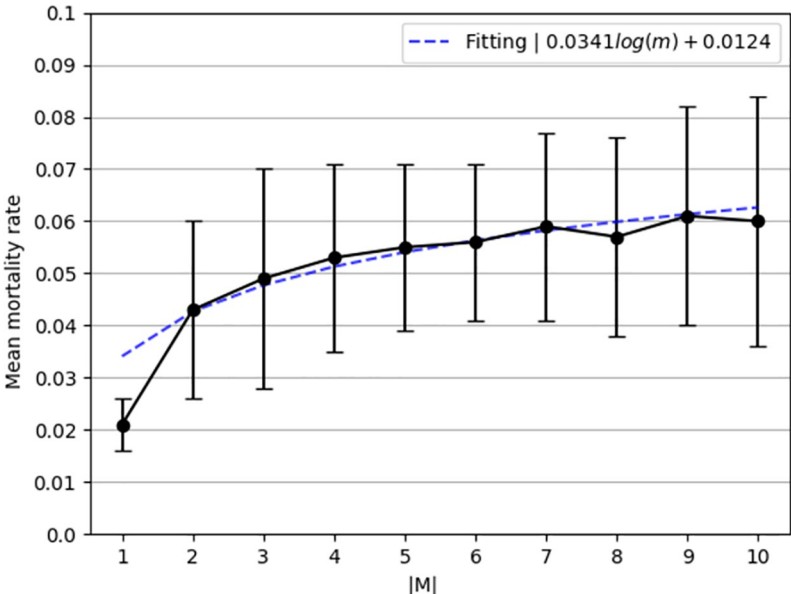

**Fig 4. Mortality rate as function of the number of strains ($|M|$).** The results are mean ± standard division for $n = 1000$ repetitions.

The *mean mortality rate* as a function of the number of strains has been computed and presented in Fig 4. Similarly, the dots are the calculated values from the simulator and the dotted line is a fitting function that is computed using the LMS with the family function $f(m) = p_1 log(m) + p_2$, resulting in

$$E[R_0](m) = 0.0341 log(m) + 0.0124. \tag{8}$$

The fitting function was obtained with a coefficient of determination $R^2 = 0.89$.

## 3 Two strain model

The two strain epidemiological model considers a constant population with a fixed number of individuals $N$. We assume a pandemic has two strains $M = \{1, 2\}$. We define a system of eight ordinary differential equations (ODEs) corresponding to eight possible epidemiological states: susceptible ($R_\emptyset$), infected by strain 1 ($R_\emptyset I_1$), infected by strain 2 ($R_\emptyset I_2$), recovered from strain 1 ($R_{\{1\}}$), recovered from strain 2 ($R_{\{2\}}$), recovered from strain 1 and infected by strain 2 ($R_{\{1\}}I_2$), recovered from strain 2 and infected by strain 1 ($R_{\{2\}}I_1$), recovered from both strains ($R_{\{1,2\}}$), and dead ($D$). The full explanation of how one obtains the model is provided in the Section 1 in S1 Appendix. A schematic transition between disease stages of an individual for the case of $|M| = 2$ is shown in Fig 5. Thus, the model for two strains is described by eight equations as

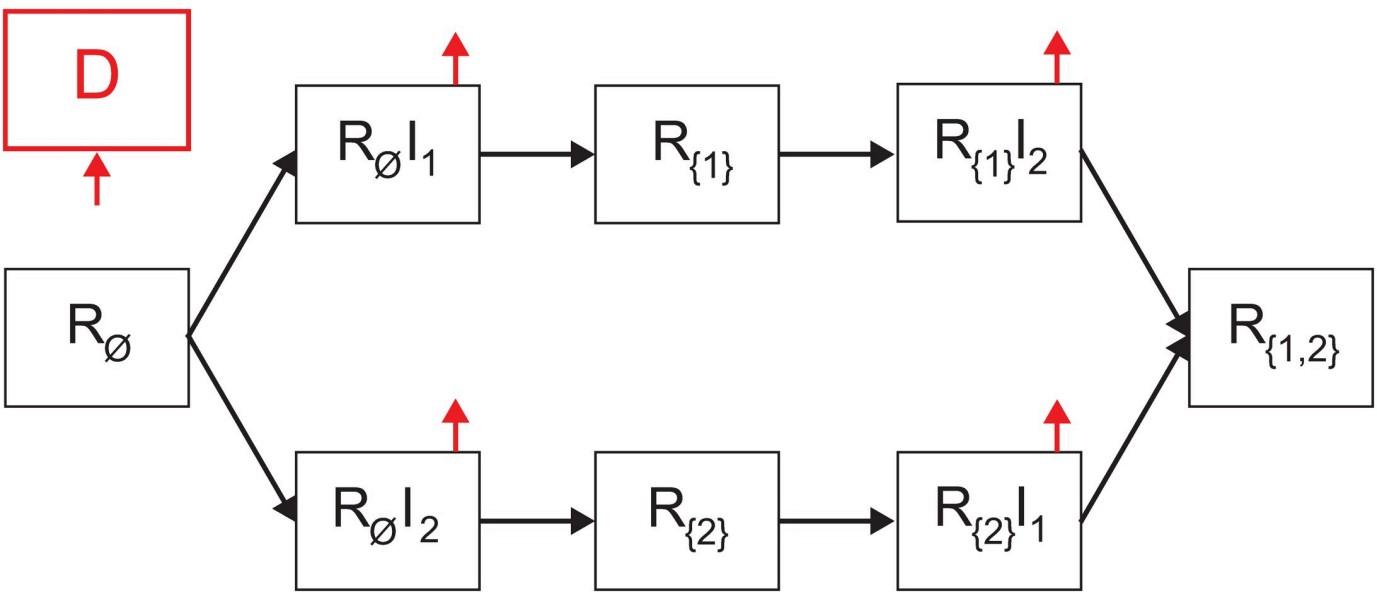

**Fig 5. Schematic view of transition between disease stages in the case where $|M| = 2$.**

follows.

$$\frac{dR_\emptyset I_1(t)}{dt} = \beta_{\emptyset,1}(R_\emptyset I_1(t) + R_{\{2\}}I_1(t))R_\emptyset(t) - \gamma_{\emptyset,1}R_\emptyset I_1(t),$$

$$\frac{dR_{\{2\}}I_1(t)}{dt} = \beta_{\{2\},1}(R_{\{2\}}I_1(t) + R_\emptyset I_1(t))R_{\{2\}}(t) - \gamma_{\{2\},1}R_{\{2\}}I_1(t),$$

$$\frac{dR_\emptyset I_2(t)}{dt} = \beta_{\emptyset,2}(R_\emptyset I_2(t) + R_{\{1\}}I_2(t))R_\emptyset(t) - \gamma_{\emptyset,2}R_\emptyset I_2(t),$$

$$\frac{dR_{\{1\}}I_2(t)}{dt} = \beta_{\{1\},2}(R_{\{1\}}I_2(t) + R_\emptyset I_2(t))R_{\{1\}}(t) - \gamma_{\{1\},2}R_{\{1\}}I_2(t),$$

$$\frac{dR_\emptyset(t)}{dt} = -R_\emptyset(t)(\beta_{\emptyset,1}(R_\emptyset I_1(t) + R_{\{2\}}I_1(t)) + \beta_{\emptyset,2}(R_\emptyset I_2(t) + R_{\{1\}}I_2(t))), \qquad (9)$$

$$\frac{dR_{\{1\}}(t)}{dt} = \gamma_{\emptyset,1}\phi_{\emptyset,1}R_\emptyset I_1(t) - \beta_{\{1\},2}(R_{\{1\}}I_2(t) + R_\emptyset I_2(t))R_{\{1\}}(t),$$

$$\frac{dR_{\{2\}}(t)}{dt} = \gamma_{\emptyset,2}\phi_{\emptyset,2}R_\emptyset I_2(t) - \beta_{\{2\},1}(R_{\{2\}}I_1(t) + R_\emptyset I_1(t))R_{\{2\}}(t),$$

$$\frac{dR_{\{1,2\}}(t)}{dt} = \gamma_{\{2\},1}\phi_{\{2\},1}R_{\{2\}}I_1(t) + \gamma_{\{1\},2}\phi_{\{1\},2}R_{\{1\}}I_2(t),$$

$$\frac{dD(t)}{dt} = \gamma_{\emptyset,1}(1 - \phi_{\emptyset,1})R_\emptyset I_1(t) + \gamma_{\{2\},1}(1 - \phi_{\{2\},1})R_{\{2\}}I_1(t)$$
$$+ \gamma_{\emptyset,2}(1 - \phi_{\emptyset,2})R_\emptyset I_2(t) + \gamma_{\{1\},2}(1 - \phi_{\{1\},2})R_{\{1\}}I_2(t).$$

The initial conditions of Eq (9) are defined for the beginning of a pandemic as follows:

$$R_\emptyset(0) = N - 2, \ R_\emptyset I_1(0) = 1, \ R_\emptyset I_2(0) = 1,$$
$$D(0) = R_{\{1\}}(0) = R_{\{2\}}(0) = R_{\{1,2\}}(0) = R_{\{1\}}I_2(0) = R_{\{2\}}I_1(0) = 0 \qquad (10)$$

Moreover,

$$N = R_\emptyset + R_\emptyset I_1 + R_\emptyset I_2 + R_{\{1\}} I_2 + R_{\{2\}} I_1 + R_{\{1\}} + R_{\{2\}} + R_{\{1,2\}} + D. \qquad (11)$$

We use a model that does not allow temporary cross-immunity and without increased susceptibility to the second infection.

For $|M| = 2$, we are interested in the equilibrium states of the model, especially stable states in which a pandemic can persist for a long time. In addition, we investigate the basic reproduction number ($R_0$), as it is an indicator of a pandemic outbreak ($R_0 > 1$), and is considered the main characteristic of a pandemic.

## 3.1 Equilibria

The equilibrium state of the model is the state in which the gradient is equal to zero [56]. Hence, Eq (12) takes the form:

$$
\begin{aligned}
&-R_\emptyset(\beta_{\emptyset,1}(R_\emptyset I_1 + R_{\{2\}} I_1) + \beta_{\emptyset,2}(R_\emptyset I_2 + R_{\{1\}} I_2)) = 0, \\
&\beta_{\emptyset,1}(R_\emptyset I_1 + R_{\{2\}} I_1)R_\emptyset - \gamma_{\emptyset,1} R_\emptyset I_1 = 0, \\
&\beta_{\emptyset,2}(R_\emptyset I_2 + R_{\{1\}} I_2)R_\emptyset - \gamma_{\emptyset,2} R_\emptyset I_2 = 0, \\
&\gamma_{\emptyset,1}\phi_{\emptyset,1} R_\emptyset I_1 - \beta_{\{1\},2}(R_{\{1\}} I_2 + R_\emptyset I_2)R_{\{1\}} = 0, \\
&\gamma_{\emptyset,2}\phi_{\emptyset,2} R_\emptyset I_2 - \beta_{\{2\},1}(R_{\{2\}} I_1 + R_\emptyset I_1)R_{\{2\}} = 0, \\
&\beta_{\{1\},2}(R_{\{1\}} I_2 + R_\emptyset I_2)R_{\{1\}} - \gamma_{\{1\},2} R_{\{1\}} I_2 = 0, \\
&\beta_{\{2\},1}(R_{\{2\}} I_1 + R_\emptyset I_1)R_{\{2\}} - \gamma_{\{2\},1} R_{\{2\}} I_1 = 0, \\
&\gamma_{\{2\},1}\phi_{\{2\},1} R_{\{2\}} I_1 + \gamma_{\{1\},2}\phi_{\{1\},2} R_{\{1\}} I_2 = 0, \\
&\gamma_{\emptyset,1}(1 - \phi_{\emptyset,1})R_\emptyset I_1 + \gamma_{\{2\},1}(1 - \phi_{\{2\},1})R_{\{2\}} I_1 \\
&+\gamma_{\emptyset,2}(1 - \phi_{\emptyset,2})R_\emptyset I_2 + \gamma_{\{1\},2}(1 - \phi_{\{1\},2})R_{\{1\}} I_2 = 0.
\end{aligned}
\qquad (12)
$$

From Eq (12), the *pandemic-free* equilibria is obtained where

$$R_\emptyset I_1^* = R_\emptyset I_2^* = R_{\{1\}} I_2^* = R_{\{2\}} I_1^* = 0,$$

since there are no more infected individuals in this state, which means all strains have gone extinct. Therefore, the equilibria states take the form:

$$R_\emptyset^* = \mu_1, \ R_{\{1\}}^* = \mu_2, \ R_{\{2\}}^* = \mu_3, \ R_{1,2}^* = \mu_4, \ D^* = N - \Sigma_{i=1}^4 \mu_i. \qquad (13)$$

According to [56], this set of states (Eq (13)) is the only asymptotically stable equilibria of the model. Nonetheless, the equilibria states where strain $i = 1$ is over are obtained where

$$R_\emptyset I_1^* = R_{\{2\}} I_1^* = 0.$$

These equilibria states are epidemiologically interesting as the extinction of one of two strains can be a turning point in multiple pandemic management policies. Thus, it is assumed (without loss of generality) that $i = 1$. Hence, from the fourth and sixth equations, one obtains that

$$RI_2^* = R_{\{1\}} I_2^* = 0.$$

Accordingly, the system converges to the *pandemic-free* equilibria.

In addition, while other equilibria states theoretically exist (by relaxing the previous assumptions), from an epidemiological point of view, the unstable equilibria are obtained in the middle of the pandemic. It is possible to see that it is enough that an individual may recover

in order to diverge from each one of these equilibria states. As such, these equilibria obtained, if any, do not provide a meaningful point in the pandemic's dynamics.

## 3.2 Basic reproduction number

The basic reproduction number, $R_0$, is defined as the expected number of secondary cases produced by a single (typical) infection in a completely susceptible population [57]. In the case of a *SIR*-based model, the basic reproduction number indicates an epidemic outbreak if $R_0 > 1$ or not if $R_0 < 1$.

To find the basic reproduction number for two strains, we use the Next Generation Matrix (NGM) approach [58]. First, we compute the new infections matrix

$$\mathbf{F} = \begin{pmatrix} \beta_{\emptyset,1} R_{\emptyset} & 0 & \beta_{\{2\},1} R_{\emptyset} & 0 \\ 0 & \beta_{\emptyset,2} R_{\emptyset} & 0 & \beta_{\{1\},2} R_{\emptyset} \\ \beta_{\{2\},1} R_{\{2\}} & 0 & \beta_{\{2\},1} R_{\{2\}} & 0 \\ 0 & \beta_{\{1\},2} R_{\{1\}} & 0 & \beta_{\{1\},2} R_{\{1\}} \end{pmatrix}. \tag{14}$$

Afterward, we compute the transfers of infections from one compartment to another matrix

$$\mathbf{V} = \begin{pmatrix} \gamma_{\emptyset,1} & 0 & 0 & 0 \\ 0 & \gamma_{\emptyset,2} & 0 & 0 \\ 0 & 0 & \gamma_{\{1\},2} & 0 \\ 0 & 0 & 0 & \gamma_{\{2\},1} \end{pmatrix} \rightarrow \mathbf{V}^{-1} = \begin{pmatrix} 1/\gamma_{\emptyset,1} & 0 & 0 & 0 \\ 0 & 1/\gamma_{\emptyset,2} & 0 & 0 \\ 0 & 0 & 1/\gamma_{\{1\},2} & 0 \\ 0 & 0 & 0 & 1/\gamma_{\{2\},1} \end{pmatrix}. \tag{15}$$

Now, $R_0$ is the dominant eigenvalue of the matrix [58].

$$\mathbf{G} = \mathbf{F}\mathbf{V}^{-1} = \begin{pmatrix} \dfrac{\beta_{\emptyset,1} R_{\emptyset}}{\gamma_{\emptyset,1}} & 0 & \dfrac{\beta_{\{2\},1} R_{\emptyset}}{\gamma_{\{1\},2}} & 0 \\ 0 & \dfrac{\beta_{\emptyset,2} R_{\emptyset}}{\gamma_{\emptyset,2}} & 0 & \dfrac{\beta_{\{1\},2} R_{\emptyset}}{\gamma_{\{2\},1}} \\ \dfrac{\beta_{\{2\},1} R_{\{2\}}}{\gamma_{\emptyset,1}} & 0 & \dfrac{\beta_{\{2\},1} R_{\{2\}}}{\gamma_{\{1\},2}} & 0 \\ 0 & \dfrac{\beta_{\{1\},2} R_{\{1\}}}{\gamma_{\emptyset,2}} & 0 & \dfrac{\beta_{\{1\},2} R_{\{1\}}}{\gamma_{\{2\},1}} \end{pmatrix} \tag{16}$$

which is obtained from the root of the representative polynomial:

$$0 = \lambda^4 - \lambda^3 \left( \frac{\beta_{\{2\},1}}{\gamma_{\{1\},2}} + \frac{\beta_{\{1\},2}}{\gamma_{\{2\},1}} + \frac{\beta_{\emptyset,1}}{\gamma_{\emptyset,1}} \right) +$$

$$\lambda^2 \left( 2\frac{\beta_{\emptyset,2}}{\gamma_{\emptyset,2}} \frac{\beta_{\{2\},1}}{\gamma_{\{1\},2}} - \frac{\beta_{\emptyset,2}}{\gamma_{\emptyset,2}} + \frac{\beta_{\{1\},2}}{\gamma_{\{2\},1}} \frac{\beta_{\emptyset,2}}{\gamma_{\emptyset,2}} - \frac{\beta_{\{1\},2}}{\gamma_{\{2\},1}} \frac{\beta_{\{1\},2}}{\gamma_{\emptyset,2}} + \frac{\beta_{\{2\},1}}{\gamma_{\{1\},2}} \frac{\beta_{\emptyset,1}}{\gamma_{\emptyset,1}} + \frac{\beta_{\{1\},2}}{\gamma_{\{2\},1}} \frac{\beta_{\emptyset,1}}{\gamma_{\emptyset,1}} - \frac{\beta_{\{2\},1}}{\gamma_{\{1\},2}} \right) +$$

$$\lambda \left( -\frac{\beta_{\emptyset,2}}{\gamma_{\emptyset,2}}^2 \frac{\beta_{\{2\},1}}{\gamma_{\{1\},2}} + \frac{\beta_{\{1\},2}}{\gamma_{\{2\},1}} \frac{\beta_{\{1\},2}}{\gamma_{\emptyset,2}} \frac{\beta_{\{2\},1}}{\gamma_{\{1\},2}} - 2\frac{\beta_{\emptyset,1}}{\gamma_{\emptyset,1}} \frac{\beta_{\emptyset,2}}{\gamma_{\emptyset,2}} \frac{\beta_{\{2\},1}}{\gamma_{\{1\},2}} + \frac{\beta_{\emptyset,1}}{\gamma_{\emptyset,1}} \frac{\beta_{\emptyset,2}}{\gamma_{\emptyset,2}} - \frac{\beta_{\emptyset,1}}{\gamma_{\emptyset,1}} \frac{\beta_{\emptyset,2}}{\gamma_{\emptyset,2}} \frac{\beta_{\{1\},2}}{\gamma_{\{2\},1}} + \tag{17}$$

$$\frac{\beta_{\emptyset,1}}{\gamma_{\emptyset,1}} \frac{\beta_{\{1\},2}}{\gamma_{\{2\},1}} \frac{\beta_{\{1\},2}}{\gamma_{\emptyset,2}} + \frac{\beta_{\{1\},2}}{\gamma_{\{2\},1}} \frac{\beta_{\{2\},1}}{\gamma_{\{1\},2}} + \frac{\beta_{\emptyset,2}}{\gamma_{\emptyset,2}} \frac{\beta_{\{2\},1}}{\gamma_{\{1\},2}} \right) +$$

$$\frac{\beta_{\emptyset,1}}{\gamma_{\emptyset,1}} \frac{\beta_{\emptyset,2}}{\gamma_{\emptyset,2}}^2 \frac{\beta_{\{2\},1}}{\gamma_{\{1\},2}} - \frac{\beta_{\emptyset,1}}{\gamma_{\emptyset,1}} \frac{\beta_{\{1\},2}}{\gamma_{\{2\},1}} \frac{\beta_{\{1\},2}}{\gamma_{\emptyset,2}} \frac{\beta_{\{2\},1}}{\gamma_{\{1\},2}} - \frac{\beta_{\emptyset,2}}{\gamma_{\emptyset,2}} \frac{\beta_{\{1\},2}}{\gamma_{\{2\},1}} + \frac{\beta_{\{2\},1}}{\gamma_{\emptyset,1}} \frac{\beta_{\{2\},1}}{\gamma_{\{1\},2}} \frac{\beta_{\{1\},2}}{\gamma_{\{2\},1}} \frac{\beta_{\{1\},2}}{\gamma_{\emptyset,2}}.$$

Using *Matlab*'s (version 2021b) symbolic programming, one is able to obtain the $R_0$. Just find the roots of the polynomial shown in Eq (17) and take the biggest one. This approach cannot be generalized for more than two strains $|M| > 2$ as the NGM will be of size $k \times k$ where $k = \Sigma_{i=1}^{|M|} nx$. Namely, the size of the NGM is larger than four and according to Galois theory [59] and based on the Abel–Ruffini theorem [60], the roots of the representative polynomial of the NGM cannot be obtained using radicals. This means one cannot provide a closed-form formula for the eigenvalues of the NGM which are used to obtain $R_0$.

## 3.3 Model validation

The model validation is divided into two phases: parameter estimation and historical fitting. The parameter estimation method allows us to use of the proposed model on a specific pandemic and the historical fitting shows the ability of the proposed model to approximate real pandemic spread dynamics given the obtained parameters.

**3.3.1 Parameter estimation.** The proposed epidemiological model parameter for the case $|M| = 2$ is obtained by fitting the proposed model onto the historical data from April 1 (2020) to December 1 (2020) of the UK by WHO [7], using the fourth-order Runge-Kutta [54] and gradient descent [61] algorithms. These dates are picked as the population in the UK during this period had not been vaccinated against the COVID-19 disease yet and a second strain (i.e., the COVID-19 UK Variant—B.1.1.7) appeared according to [62], which based their analysis on clinical testing and later reverse engineering of the mutation's appearance [63]. Both point to the same period even though there is no full agreement on the specific dates of the appearance of the mutation. Specifically, we randomly guess the values of the model's parameters, solving the system of ODEs using the fourth-order Runge-Kutta method and computing the Gaussian ($L_2$) distance from the historical data. In particular, we used the daily number of infection, recovered, and deceased individuals. As such, the fitness function takes the form:

$$F(H, P)_{[t_0, t_f]} := \sqrt{\Sigma_{t=t_0}^{t_f} \left( (H[S](t) - P[S](t))^2 + (H[R](t) - P[R](t))^2 + (H[I](t) - P[I](t))^2 \right)}, \tag{18}$$

where $H[X](t)$ is the historical size of the population at the epidemiological state $X$ at time $t$ and $P[X](t)$ is the model's prediction size of the population at the epidemiological state $X$ at time $t$. The model's $P[I]$ and $P[R]$ refer to all states for the form $R_j I_i$ and $R_j$, respectively.

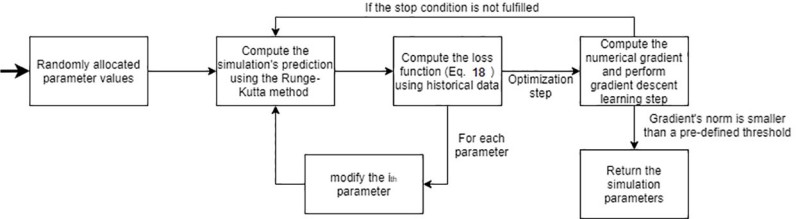

**Fig 6. A schematic view of the fitting method.**

Afterward, we repeated this process while modifying the value of a single parameter by some pre-defined value $\delta = 0.01$, obtaining a numerical gradient. At this stage, we used the gradient descent algorithm in order to find the values that minimize the model's $L_2$ distance from the historical data using Eq (18). The process is stopped once the gradient's ($L_1$) norm is smaller than some pre-defined threshold value $\epsilon = 0.1$. The entire process is repeated $r = 100$ times and the parameter values that are obtained most often are decided to be the model's parameter value. The values for ($\delta, \epsilon, r$) are manually picked. A schematic view of the fitting method is presented in Fig 6.

**3.3.2 Historical fitting.** In order to numerically evaluate the ability of the proposed model to fit real epidemiological data, we decided to simulate the COVID-19 pandemic in the United Kingdom (UK). This case is chosen due to the availability of epidemiological data and since a COVID-19 strain is known to originate in the UK [7, 64]. Therefore, we computed the parameter values, assuming the initial conditions taking the form:

$$R_{\varnothing}(0) = 67200000, \ R_{\varnothing}I_1(0) = 100, \ R_{\varnothing}I_2(0) = 1, \ D(0) = 0$$

$$\tag{19}$$

$$R_{\{1\}}(0) = \ R_{\{2\}}(0) = R_{\{1,2\}}(0) = R_{\{1\}}I_2(0) = R_{\{2\}}I_1(0) = 0.$$

where $R_{\varnothing}(0) = 67200000$ to represent the size of the UK population in the beginning of the pandemic. A summary of the obtained parameter values is shown in Table 1, such that 27% of the random parameter value initial conditions converged to the values with $d_{L_2} = 0.089$. Namely, the model has a daily mean square error of 8.9%.

One needs to be cautious with this historical fitting of COVID-19 data due to historical error in COVID-19 related death classification, undersampling of infected individuals, and errors associated in identifying the strain individuals are infected with [65, 66]. These errors may result in off representation of the epidemiological dynamics and as such wrong parameter values. Nonetheless, COVID-19 is the most documented pandemic in history [67] and therefore it is the best candidate to use despite the problems associated with it.

A fitting dynamics between the historical data (circle, black) and the model's prediction (axes, blue) is shown in Fig 7, where the x-axis describes the time from September 1 (2020) to December 1 (2020), and the y-axis describes the daily basic reproduction number ($R_0$). The historical basic reproduction number ($R_0$) from WHO is computed using the following formula $R_0(t) := \frac{I(t+1)-I(t)}{R(t+1)-R(t)}$ ..

# 4 Discussion

We have developed a mathematical model and a computer simulation aiming at establishing the connections between the number of pandemic disease strains and the pandemic's spread

**Table 1. A summary of the model parameters and values for the case of |M| = 2, obtained from the fitting process to the historical WHO COVID-19 data from April 1 (2020) to December 1 (2020).**

| Parameter Definition | Symbol | Value |
|---|---|---|
| Infection rate of the strain ($i = 1$) [1] | $\beta_{\emptyset,1}$ | 0.04 |
| Infection rate of the strain ($i = 2$) [1] | $\beta_{\emptyset,2}$ | 0.07 |
| Infection rate of the strain ($i = 2$), after recovery from the strain ($i = 1$) [1] | $\beta_{\{1\},2}$ | 0.01 |
| Infection rate of the strain ($i = 1$), after recovery from the strain ($i = 2$) [1] | $\beta_{\{2\},1}$ | 0.02 |
| The average duration that it takes for an individual to recover from the strain ($i = 1$) in days [$t^{-1}$] | $\gamma_{\emptyset,1}$ | 0.08 |
| The average duration that it takes for an individual to recover from the strain ($i = 2$) in days [$t^{-1}$] | $\gamma_{\emptyset,2}$ | 0.06 |
| The average duration that it takes for an individual to recover from the strain ($i = 1$) after recovering from the strain ($i = 2$) in days [$t^{-1}$] | $\gamma_{\{2\},1}$ | 0.21 |
| The average duration that it takes for an individual to recover from the strain ($i = 2$) after recovering from the strain ($i = 1$) in days [$t^{-1}$] | $\gamma_{\{1\},2}$ | 0.17 |
| The probability an infected individual will recover from the strain ($i = 1$) [1] | $\phi_{\emptyset,1}$ | 0.98 |
| The probability an infected individual will recover from the strain ($i = 2$) [1] | $\phi_{\emptyset,2}$ | 0.96 |
| The probability an infected individual will recover from the strain ($i = 2$) after recovering from the strain ($i = 2$) [1] | $\phi_{\{1\},2}$ | 0.99 |
| The probability an infected individual will recover from the strain ($i = 3$) after recovering from the strain ($i = 1$) [1] | $\phi_{\{2\},1}$ | 0.99 |

in the population for any pathogen, under the epidemiological *SIRD* model. Unlike the previous modeling approaches [12, 38, 40], we have extended the strain diversity for any arbitrary number ($m$) and did not introduce any pathogen-specific attributes, keeping the model as generic as possible.

We have shown that for the case of only two strains (e.g., |M| = 2), the only stable equilibria states are when the pandemic is over for both strains ($R_{\emptyset}I_1^* = R_{\emptyset}I_2^* = R_{\{1\}}I_2^* = R_{\{2\}}I_1^* = 0$), as shown in Section 3.3.2. The result of the equilibrium analysis is that the *pandemic-free* states

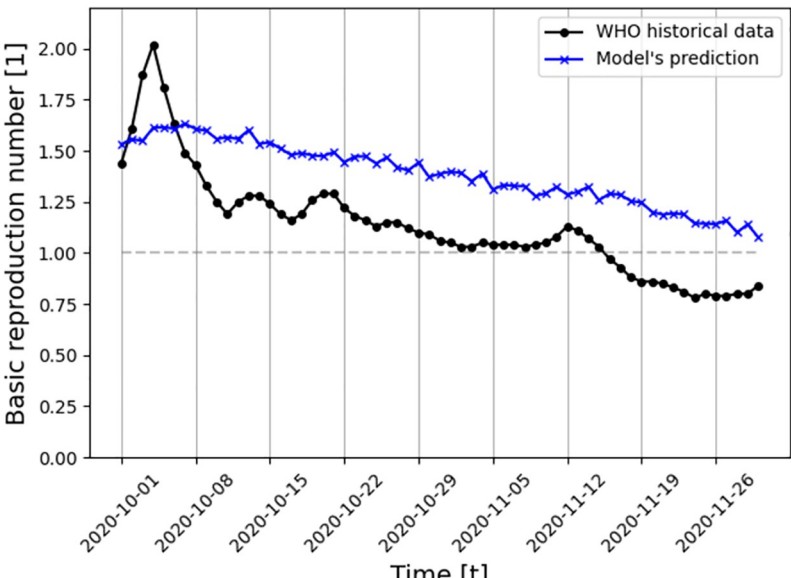

**Fig 7. Daily $R_0$ in UK between September 1 and December 1 (2020) comparison between the historical data (specifically, the daily number of infected, recovered, and dead individuals) and the proposed model predictions (for |M| = 2).** The gray horizontal line indicates $R_0 = 1$. The model's parameter values are shown in Table 1.

are stable only when the epidemics of the two strains cease; that is, after the end of the general pandemic (Eq (13)).

Moreover, an analytical computation of the basic reproduction number ($R_0$) requires information on infections between individuals with different strains, which is not realistically available. Therefore, an immediate result of the model is that once a pandemic developed secondary strains, a numerical and statistical approximation of $R_0$ is left to be the only feasible approach.

In addition, the proposed model is evaluated on the COVID-19 pandemic (for the case of the UK) and has shown promising ability to fit a long period of multi-strain historical data (eight months, 8.9% daily mean square error). A prediction of the last two months of this period is shown in Fig 7, based on the obtained model's parameter values which are presented in Table 1. Strain $i = 1$ is mapped to the original strain of COVID-19. Since at the beginning of the pandemic, this was only a single strain, the measured epidemiological values are necessarily associated with this strain. This is not the case for measurements of periods where two or more strains existed. The proposed model captures a general trend of decreasing $R_0$ during this period while not matching the data closely as it intentionally does not take into consideration other social and epidemiological dynamics, allowing analytical analysis to be considered. However, future extensions of the proposed model should be able to predict more closely historical pandemic events.

According to Voinsky et al. [68], the average recovery rate of strain ($i = 1$) is 0.0714 while the model predicted $\gamma_{\emptyset,1} = 0.08$ (where the approximation size is $\delta = 0.01$), as presented in Table 1. In addition, according to WHO [7], the average mortality rate of this period is $\sim 0.0138$ while the model predicted that the average mortality rate from this strain is $1 - \phi_{\emptyset,1}$ = 0.02. Thus, while the model is simple, it is able to capture the biological and epidemiological properties of the pandemic.

Furthermore, we evaluated the influence of the number of strains on the mean basic reproduction number ($E[R_0]$), mortality rate, and a maximum number of infected individuals, as shown in Figs 2, 4 and 3, respectively. We show that the basic reproduction number is upper bounded by taking into consideration only the most aggressive strain. Formally, we computed a one-sided confidence interval between the baseline and the most aggressive strain dynamics and found that 0 is not included in the confidence interval ($\alpha = 0.05$). Hence, we conclude that the two dynamics are different and that the most aggressive strain dynamic is an upper bound of the baseline dynamics. This result agrees with the one obtained by [40] for arbitrary number of mutations and [42] for the case of $|M| = 2$. In particular, [42] has shown that analytically the most aggressive strain and the baseline dynamic converge which is shown in Fig 2. The slight difference in the mean value is associated with the stochastic nature of the numerical computation method. An immediate outcome is that the proposed model is upper bounded by the *SIRD* model with the slight modification that each individual can be infected up to $|M|$ times. This means one can get a statistically similar result (on average) to a pandemic with $|M|$ strains by using a simpler model that requires less biological and epidemiological data compared to the proposed model. These results agree with the analysis performed by Dang et al. [36] on a multi-strain model for influenza.

Based on Eq (7), the maximum number of infected individuals is growing in a logarithmic manner to the number of strains when the latter occurs simultaneously. In a similar manner, based on Eq (8), the mortality rate is growing in a logarithmic manner to the number of strains when the latter occur simultaneously. We numerically show in Figs 3 and 4 that the epidemiological properties which indicate the severity of the pandemic in a well-mixed population grow in a logarithmic manner as a function of the number of strains ($|M|$). This connection indicates that the first few strains make a relatively large contribution to the mortality and pandemic

spread dynamics, but as the number of strains grows, each strain contributes less to these numbers. Policymakers can take advantage of this link when planning intervention policies to contain the spread of a pandemic, given that new strains can emerge during pathogen mutation. The code developed for this model is publicly available as open-source.

Several possible future research directions emerge from the proposed initial modeling. First, one can introduce a fixed delay parameter to the occurrence of strains, investigating the influence of this parameter on the epidemiological spread similar to the model proposed by Arruda et al. [40]. Second, one can take into consideration more detailed biological settings, assuming the stochastic occurrence of the strains from some distribution. Third, one can allow reinfection of the same strain, extending the proposed model to a *SIRS*-based model. These directions aim to better represent a real pandemic where several strains do not exist from the beginning of the pandemic. Moreover, one can introduce a similarity matrix between the strains as they are mutations of an original strain, which are reflected by the immunity response to reinfection of different strains or a cross-immunity response as proposed by [69]. In the same direction, adding an Exposed state would make the proposed model more biologically accurate, since most strains have an incubation period before the host becomes infectious. The multi-strain model is a theoretical platform that will help guide the decision-making process in the event of a pandemic crisis while providing the forecast of the results of the selected course of action.

## Supporting information

**S1 Appendix. Single and two mutations model.**
(PDF)

## Author Contributions

**Formal analysis:** Teddy Lazebnik.

**Investigation:** Teddy Lazebnik.

**Methodology:** Teddy Lazebnik.

**Project administration:** Teddy Lazebnik, Svetlana Bunimovich-Mendrazitsky.

**Software:** Teddy Lazebnik.

**Supervision:** Svetlana Bunimovich-Mendrazitsky.

**Validation:** Teddy Lazebnik, Svetlana Bunimovich-Mendrazitsky.

**Visualization:** Teddy Lazebnik.

**Writing – original draft:** Teddy Lazebnik.

**Writing – review & editing:** Svetlana Bunimovich-Mendrazitsky.

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
