## [Decision Letter · Decision Letter 0]

7 Jan 2022

PONE-D-21-35947Generic Approach For Mathematical Model of Multi-Strain PandemicsPLOS ONE

Dear Dr. Lazebnik,

Thank you for submitting your manuscript to PLOS ONE. After careful consideration, we feel that it has merit but does not fully meet PLOS ONE’s publication criteria as it currently stands. Therefore, we invite you to submit a revised version of the manuscript that addresses the points raised during the review process.

We look forward to receiving your revised manuscript.

Kind regards,

Martial L Ndeffo Mbah, Ph.D

Academic Editor

PLOS ONE

Journal Requirements:

 [The funders had no role in study design, data collection and analysis, decision to publish, or preparation of the manuscript.]

4. One of the noted authors is a group or consortium [Lorem Ipsum

Consortium]. In addition to naming the author group, please list the individual authors and affiliations within this group in the acknowledgments section of your manuscript. Please also indicate clearly a lead author for this group along with a contact email address.

Additional Editor Comments:

The reviewers have raised important questions that need to be thoroughly addressed before that manuscript can be reconsidered for publication. The revised version of the manuscript will be send back to reviewers for further evaluation before an editorial decision is made.

Reviewers' comments:

Reviewer's Responses to Questions

**Comments to the Author**

1. Is the manuscript technically sound, and do the data support the conclusions?

Reviewer #1: No

Reviewer #2: Yes

2. Has the statistical analysis been performed appropriately and rigorously? 

Reviewer #1: No

Reviewer #2: No

3. Have the authors made all data underlying the findings in their manuscript fully available?

Reviewer #1: Yes

Reviewer #2: Yes

4. Is the manuscript presented in an intelligible fashion and written in standard English?

Reviewer #1: Yes

Reviewer #2: Yes

5. Review Comments to the Author

Reviewer #1: The authors present a multi-strain epidemic model that, albeit quite complicated, does not address basic issues in an epidemic: such as reinfection and latency periods, which for example play a foremost role in the current COVID-19 epidemic. These issues are already considered in a recent PloS One publication of which the authors seem unaware; hence, an updated literature review is recommended.

The authors seem to be a bit late with their model, as this journal already published a multi-strain epidemic model recently (https://doi.org/10.1371/journal.pone.0257512). Observe that the model in the paper has a straightforward calculation of the reproduction number which essentially arrives at the same conclusion as your paper: that the reproduction number is ruled by the single strain with the highest reproduction number. Hence, that model seems richer and more useful as an off-shelf tool to predict epidemics, whilst also including optimal control.

All in all, the paper seems overly complicated whilst not tackling important epidemic issues. The innovations not enough to warrant publication in the current form.

Minor comments:

- multi-strain is an adjective, not a noun as used in the abstract

- there are many other papers regarding multi-strain models and optimal control which were not explored in the literature review

- experimentation with COVID data should be addressed carefully, as the data is non-standard and depends upon testing strategies and government reports, which may vary and often are way off the mark. Hence, you may be adjusting your model to a reality that does not exist: https://doi.org/10.1016/S0140-6736(21)00183-5

https://www.bmj.com/content/373/bmj.n1442

Reviewer #2: Introduction: There has been a lot of work on multi-strain pandemics, many of which you have cited in the paper. Mainly Khayar and Allali for the COVID-19 pandemic. Fudolig & Howard (PLoS One, 2020) also studied a multi-strain SIR model with selective immunity through vaccination. The authors did not provide enough justification on why their work is needed. Why do we want to generalize it to M strains when it is expected that only one or two strains will be dominant in the population anyway? How does your model detect sharp changes in dynamics due to pandemic modifications (line 65)? Please emphasize it more in the revised manuscript.

Line 89: Chine -> China

Section 2.1:

-Can you cite the source for the mean basic reproduction number, mortality rate, and maximum number of infectious? Were these properties that you defined on your own? How does the mean basic reproduction number agree with the first generation matrix approach by van den Driesche and Watmough?

-Most aggressive strain: Is this an L^3 norm? Why did you use this metric? Please elaborate.

2.2:

-Line 205 says that you assumed that no individuals recovered or died as a result of the pandemic. Then why do you have a non-zero mean mortality rate? Shouldn't this mean that your D compartment should be zero at all times?

-Fudolig &Howard determined that the reproduction number of a two-strain SIR epidemic is the reproduction number of the most aggressive strain (max R_0), so it is expected for the two curves to be close. The slightly lower baseline value is only because of the averaging.

-What is the point of modeling the curves using a logarithmic function? What does that information tell you? In Figure 3, how do you explain the sudden rise for M=5,6?

Section 3

-Fig. 4: Mean mortality rate graph when the authors assumed that there are no individuals who died as a result of the pandemic. If it was just a typo, how would you explain this rising mortality rate? Is it dominated by the deadliest strain (highest death rate on its own?)

-Lines 265-268: The unstable equilibria provide meaning especially in multi-strain epidemics when there could be a change in dominant strain in the population. In light of recent events, it can explain the shift between an equilibrium dominated by the COVID-19 Delta variant and the COVID-19 Omicron variant. It is important to look at the stability conditions where only one strain survives.

-Section 3.2: Can you still find the eigenvalues of the NGM for M>2 numerically? If not, this might be a good explanation to why you used the equation for the basic reproduction number in Section 2.1.

Section 4:

-Figure 7: The model prediction overestimates the reproduction number after a certain time. Given how your model overestimates the WHO historical data, how useful is your model now in predicting multiple strain SIR models?

-Line 374: Where is this t-test? Which processes were compared?

-If you used i=1 for COVID, how about i=2? Isn't the crux of the paper to be able to account for the other variants? How is this different compared to other single-strain SIR models that are more accurate in calculating the reproduction number.

Formatting errors:

-I do not think that the authors worked with "the Loren Ipsum Consortium". Please revise the template file.

-page 16: footnote "add here later" for the open source code should be revised.

6. PLOS authors have the option to publish the peer review history of their article (what does this mean?). If published, this will include your full peer review and any attached files.

Reviewer #1: No

Reviewer #2: No

---

## [Author Response · Author response to Decision Letter 0]

15 Jan 2022

First of all, we would like to thank the editor and the reviews for the careful review and the thought-provoking comments. 

Editor

Comment 1: “Please ensure that your manuscript meets PLOS ONE's style requirements, including those for file naming. The PLOS ONE style templates can be found at “

Answer 1: We used PLOS ONE’s style guide, we hope now it is fine now.

Comment 2: “Please include your amended statements within your cover letter; we will change the online submission form on your behalf.”

Answer 2: We add the “The authors received no specific funding for this work.” statement to the cover letter.

Comment 3: “We note that you have stated that you will provide repository information for your data at acceptance.”

Answer 3: All the relevant data used in this research is publicly available - we cited the sources in the manuscript. 

Comment 4: “One of the noted authors is a group or consortium [Lorem Ipsum Consortium]. “

Answer 4: We apologize, this is just a technical mistake due to the placeholder in the template. Of course, there is no author “Lorem Ipsum Consortium”. 

Comment 5: “ Please include captions for your Supporting Information files at the end of your manuscript, and update any in-text citations to match accordingly. “

Answer 5: We include the supporting information files at the end of the manuscript.

Reviewer #1

Comment 1: “The authors present a multi-strain epidemic model that, albeit quite complicated, does not address basic issues in an epidemic: such as reinfection and latency periods, which for example play a foremost role in the current COVID-19 epidemic.”

Answer 1: The reviewer points out important epidemiological processes that are missing from the model. Indeed, the proposed model does not address the latency periods (commonly known as the exposed state) since the model, as stated by the reviewer, is complex already, and adding Expose State introduces an additional level of complexity that we were trying to avoid. Regarding the reinfection, the model allows an individual in the population to be reinfected by other strains of the pandemic, as shown in figure 1. Other models allow reinfection of the same strain such as the SIS model (Reluga, 2008) but our model is based on the SIR model, which is a valid epidemiological model, used in an endless number of works (Oka et al., 2020; Cooper et al., 2020; Khyar and Allali, 2020), obtaining wonderful results on a wide range of epidemiological scenarios. Moreover, reinfection from the same strain is valid for the COVID-19 but not for all types of pandemics. For example, HIV is considered a pandemic but does not have reinfection (Quinn, T. C. Global burden of the HIV pandemic. Lancet, 1996.) of the same mutation. HIV is a valid example is in this context as it is multi-strain due to the high mutation rate (Cuevas et al., Extremely High Mutation Rate of HIV-1 In Vivo. Plos Biology, 2015.). Since the proposed model is generic and not focused on COVID-19 only, it is not absolute that reinfection is even occurring. Hence, the latency periods and reinfection of the same strain are explicitly decided to not be included in the current model in order to allow it to include as many pathogens as possible since the main aim of this work is to propose a rigorous mathematical framework for multiple strains and not the best, most accurate, model for a single pathogen. That said, we would explore these suggestions in follow-up work.

Comment 2: “These issues are already considered in a recent PloS One publication of which the authors seem unaware; hence, an updated literature review is recommended.”

Answer 2: Thank you for the reference to the recent publication. We introduce new related works, including the paper (Edilson et al. 2021) proposed by the reviewer and one proposed by the second reviewer because it is very important for our research. Unfortunately, we originally submitted our manuscript before the referred works have been published and therefore were not able to address them...

Comment 3: “Observe that the model in the paper [added by the authors: (Edilson et al., 2021)] has a straightforward calculation of the reproduction number which essentially arrives at the same conclusion as your paper: that the reproduction number is ruled by the single strain with the highest reproduction number. Hence, that model seems richer and more useful as an off-shelf tool to predict epidemics, whilst also including optimal control.”

Answer 3: Thank you for the reference. Indeed, the works are similar in nature and it is gladdening that both Edilson et al. and us obtain a similar analytical result since this shows the robustness of the outcome in different modeling attempts. Nevertheless, the paper by Edilson et al. (2021) models dynamics in which there are n strains focused on pathogens with reinfection dynamics (please see comment #1). We would like to point out that their main difference between the two models: the proposed model takes into consideration the order of infection in the individual level allowing a complex immunity dynamic. For instance, in our model, infection in strain 1 and then strain 2 is different than strain 2 and then strain 1 in infection, recovery, and death rates. This addition generalized the SIR model in a different way than Edilson et al. and therefore richer in other cases.

Regarding the optimal control comment. Indeed, a large body of work aims to compute optimal control with a wide range of intervention protocols such as masks, lockdown, social distance and etc. While a question of interest, we believe it is out of scope for the current work since it aims to provide a realistic mathematical framework that later can be used to obtain optimal control. This claim mainly relies on the fact that the proposed model aims to be generic and an intervention policy that is appropriate to one pandemic (for instance mask-wearing for air-borne pathogens like COVID-19) will not hold for others (like HIV which requires pair-wise contact).

Comment 4: “multi-strain is an adjective, not a noun as used in the abstract.”

Answer 4: Thank you, we rephrased the abstract accordingly.

Comment 5: “there are many other papers regarding multi-strain models and optimal control which were not explored in the literature review”

Answer 5: Following this comment and other comments, we updated the introduction section, introducing four new works related to multi-strain pandemic models.

Comment 6: “experimentation with COVID data should be addressed carefully, as the data is non-standard and depends upon testing strategies and government reports, which may vary and often are way off the mark. Hence, you may be adjusting your model to a reality that does not exist”

Answer 6: Thank you for pointing it out and we completely agree with this comment. Indeed, a large number of works have been published on the subject providing a range of statistical and numerical methods to overcome this error. However, neither these works nor we are able to provide a widely-agreed baseline for the COVID-19 numbers. Moreover, a critique of the WHO historical data has been heard multiple times in the context of several governments providing inaccurate values and problematic sampling methods of the populations. Nonetheless, the WHO data is still considered a consensus on the COVID-19 and is widely used by researchers and decision-makers. It will be hard to find someone that will not agree that the COVID-19 pandemic is the most documented, measured, and open access in recorded history. Hence, it is only entreated to use this data despite the problems associated with it. To summarize, we agree with the comment, and following this comment, we better express the challenges with using COVID-19 data in Section 3.2.2.

Reviewer #2

Comment 1: “Fudolig & Howard (PLoS One, 2020) also studied a multi-strain SIR model with selective immunity through vaccination. ”

Answer 1: We thank the reviewer for bringing the newly published work to our attention. The work of Fudolig & Howard (2020) impressively explores two-strains dynamic and it enriched the review of works on multi-strain’s research in Section 1 (Introduction).

Comment 2: “The authors did not provide enough justification on why their work is needed”

Answer 2: Thank you for this comment. We evaluate the contribution of the proposed model in the Introduction section, second paragraph from the end. 

Comment 3: “Why do we want to generalize it to M strains when it is expected that only one or two strains will be dominant in the population anyway?”

Answer 3: Following this comment, we better convey this manner in the Discussion section. We would like to shortly address this comment here as well: the motivation is two-folded. First, this is the first step in a way for a generalized SIR model with multiple strains that interchange with each other. The following step, which we are currently exploring, is replacing the strains with mutations - making it even more realistic as one can observe from the one currently taking place situation worldwide with the 4th or even 5th mutations of COVID-19. This dynamic is of course, does not unique just to COIVD-19 as other pathogens standing in the heart of other pandemics have a mutation process. For instance, HIV (Cuevas et al., Extremely High Mutation Rate of HIV-1 In Vivo. Plos Biology, 2015) or Influenza (Minayev, P., Ferguson, N., Improving the realism of deterministic multi-strain models: implications for modeling influenza A. Journal of the Royal Society Interface, 2008). Second, while pandemics are associated with pathogens, the SIR model is commonly used in rumor spread dynamics (https://www.aimsciences.org/article/doi/10.3934/dcdsb.2020124) and other network-based dynamics. For example, multiple rumors occurring at the same time and other similar scenarios can make use of such a mathematical framework.

Comment 4: “How does your model detect sharp changes in dynamics due to pandemic modifications (line 65)? Please emphasize it more in the revised manuscript.”

Answer 4: Thank you for pointing it out. After a careful review, we removed this statement from the manuscript as indeed we do not provide numerical or analytical results supporting this claim. 

Comment 5: “Can you cite the source for the mean basic reproduction number, mortality rate, and a maximum number of infectious? Were these properties that you defined on your own? ”

Answer 5: Thank you for the suggestion, we cite relevant manuscripts in Section 2.1, including (Breda et al. 2021; Chatterjee et al. 2020; Baud et al. 2020; Lazebnik et al. 2021). 

Comment 6: “How does the mean basic reproduction number agree with the first generation matrix approach by van den Driesche and Watmough?”

Answer 6: The proposed definition for the basic reproduction number is a forward-Euler numerical approximation to the basic reproduction number one obtains from the next generation matrix approach by van den Driesche and Watmough. The reason we use this formalization and not the one obtains from the next matrix approach is that it requires a simple computation for a specific simulation while the next-generation matrix provides a more analytical result for the dynamics and therefore is very sensitive to the model’s parameters over long periods of time such in the proposed simulations according to (Breda, 2021). 

Comment 7: “Most aggressive strain: Is this an L^3 norm? ”

Answer 7: Yes, indeed. We stated it in the paper, in section 2.1.

Comment 8: “Why did you use this metric? Please elaborate.”

Answer 8: We apologize that the definition was omitted in the former version. Following this comment, we introduce in Section 2.2 a more detailed explanation for this definition. The motivation of this metric is that a higher infection rate, longer recovery rate, and higher death rate are associated with a more aggressive strain. However, due to the complexity of the pandemic spread dynamics, it is not straightforward which one of these properties is more important if any, and therefore the comparison between two strains is performed on the three properties simultaneously.

Comment 9: “Line 205 says that you assumed that no individuals recovered or died as a result of the pandemic. Then why do you have a non-zero mean mortality rate? Shouldn't this mean that your D compartment should be zero at all times?”

Answer 9: You are right - It seems to be bad wording. We mean that for the initials condition, we assume that there are no recovered or dead individuals from the pandemic as the initial condition should take place in its beginning. This statement is not referring to the pandemic dynamics. We rephrased this statement to better convey this point and now it is written like this: “ In addition, it is assumed that no individuals have recovered or died yet due to the pandemic at the beginning of the pandemic”. 

Comment 10: “Fudolig &Howard determined that the reproduction number of a two-strain SIR epidemic is the reproduction number of the most aggressive strain (max R_0), so it is expected for the two curves to be close. The slightly lower baseline value is only because of the averaging.”

Answer 10: Thank you for pointing it out. We introduce this explanation and extend it in the Discussion section.

Comment 11: “What is the point of modeling the curves using a logarithmic function? What does that information tell you? ”

Answer 11: The idea of fitting the dynamics is to extract a functional rule for the dynamics between the number of strains and each one of the epidemiological properties. The logarithmic function is a simple function that obtains good fitting results (R^2 = 0.79 and 0.89 for the mean basic reproduction number and the mortality rate). The logarithmic function indicates that the introduction of new strains has a reducing effect on these epidemiological properties which can be of interest to decision-makers. The fitting to the logarithmic function by itself (using the least mean square method) is not interesting but the outcome that the introduction of new mutation has a lower contribution to these epidemiological properties is of interest. Following this comment, we better convey this idea in the Discussion section and we thank the reviewer for it.

Comment 12: “In Figure 3, how do you explain the sudden rise for M=5,6?”

Answer 12: The sudden rise for M=5,6 may seem to be a bit off. Figure 3 obtained numerically and since the system is very noisy due to the large number of stochastic processes it simulates, we think the sudden rise for M=5,6 is associated with the statistical error, as one can notice by the large STD of the values. Another indication of the numerical error is the relative fitting of the logarithmic function with a coefficient of determination of 0.79. Following this comment, we aimed to better convey this point in the Discussion section. 

Comment 13: “Fig. 4: Mean mortality rate graph when the authors assumed that there are no individuals who died as a result of the pandemic. If it was just a typo, how would you explain this rising mortality rate? Is it dominated by the deadliest strain (highest death rate on its own?)”

Answer 13: Thank you for drawing our attention to this abnormality. Following comment #9, it was bad wording as of course the model assumes mortality (shown both in Eqs. (1-3), Figure 1, and the numerical analysis). Therefore, a mean mortality rate graph is an entreated analysis of the proposed model. 

The rising in mortality rate can be associated with the larger number of strains and therefore the larger times one can be infected and die. Moreover, as shown by Figure 3 there are just more infected individuals so there are more cases in which individuals can die. Nonetheless, these phenomena are limited, and the commelative effected reducing in the introduction of new strains - resulting in the logarithmic increasement Following this comment, we introduce these explanations to the Discussion section.

Comment 14: “Lines 265-268: The unstable equilibria provide meaning especially in multi-strain epidemics when there could be a change in dominant strain in the population. In light of recent events, it can explain the shift between an equilibrium dominated by the COVID-19 Delta variant and the COVID-19 Omicron variant. It is important to look at the stability conditions where only one strain survives.”

Answer 14: Thank you for this comment. We also found this idea interesting and tried to tackle it. However, at the moment, we think the proposed model is not taking into consideration several properties of the COVID-19 pandemic like age-groups (children, adults, elderly), exposed phase, asymptomatic and symptomatic infected individuals, and much more to properly provide an explanation to the proposed query. Nonetheless, this idea is of high interest and we plan to try and tackle it in the near future with the follow-up models based on this one. 

Comment 15: “Section 3.2: Can you still find the eigenvalues of the NGM for M>2 numerically? If not, this might be a good explanation to why you used the equation for the basic reproduction number in Section 2.1”

Answer 15: You are right. Below Eq. (17) we state that one cannot find using radicals (namely, via a close formula) the eigenvalues of the NGM for |M|>2. One can find the eigenvalues using either numerical methods for polynomial’s roots finding or eigenvalue computation such as (Lanczos, 1950) and (Trefethen and Bau, 1997). However, these methods are unstable for large polynomials and matrices which may result in large computation errors. Moreover, these methods can be highly time- and resource- consuming for a large number of strains. 

Comment 16: “Figure 7: The model prediction overestimates the reproduction number after a certain time. Given how your model overestimates the WHO historical data, how useful is your model now in predicting multiple strain SIR models?”

Answer 16: Since the proposed model does not take into consideration pandemic intervention policies (i.e., lockdowns, vaccinations, etc.) we were not able to find COVID-19 related data that contains multiple strains on the one hand but does not contain pandemic intervention policies on the other hand. As such, we argue an experiment with the proposed model will not be fair. In order to tackle this challenge, in the follow-up work, we introduce a historical fitting that takes into consideration several pandemic intervention policies, which will allow us to conduct this experiment and declare the results

Comment 17: “Line 374: Where is this t-test? Which processes were compared?”

Answer 17: Thank you for pointing our attention to this - we rephrased this statement, stating the missing information and now it is written like this: “Formally, we perform a paired two-tail T-test between the baseline and the most aggressive strain dynamics in order to evaluate if the processes differ in a statistically significant way with \\(\\alpha = 0.05\\) and obtain that 0 is not in the confidence interval of the statistical test”.

Comment 18: “-If you used i=1 for COVID, how about i=2? Isn't the crux of the paper to be able to account for the other variants? How is this different compared to other single-strain SIR models that are more accurate in calculating the reproduction number.”

Answer 18: Similar to Edilson et al. (2021), we treat the two mutations of the COVID-19 as two different strains. As such, the experiment de-facto includes two strains. 

Comment 19: “I do not think that the authors worked with "the Loren Ipsum Consortium". Please revise the template file.”

Answer 19: Thank you for this comment, this is indeed a technical mistake. We removed the “Loren Ipsum Consortium” text from the paper.

Comment 20: “page 16: footnote "add here later" for the open-source code should be revised.”

Answer 20: Fixed. We introduce the code with the other supporting information according to the journal’s requirements.

---

## [Decision Letter · Decision Letter 1]

9 Feb 2022

PONE-D-21-35947R1Generic Approach For Mathematical Model of Multi-Strain PandemicsPLOS ONE

Dear Dr. Lazebnik,

Thank you for submitting your manuscript to PLOS ONE. After careful consideration, we feel that it has merit but does not fully meet PLOS ONE’s publication criteria as it currently stands. Therefore, we invite you to submit a revised version of the manuscript that addresses the points raised during the review process.

We look forward to receiving your revised manuscript.

Kind regards,

Martial L Ndeffo Mbah, Ph.D

Academic Editor

PLOS ONE

Journal Requirements:

Additional Editor Comments (if provided):

Thank you for submitting a revised version of your manuscript. Reviewers have raised additional comments which should be addressed before the manuscript can be deemed suitable for publication. Please, focus on addressing reviewer #2 comment and the grammatical errors highlighted by reviewer #1.

Reviewers' comments:

Reviewer's Responses to Questions

**Comments to the Author**

1. If the authors have adequately addressed your comments raised in a previous round of review and you feel that this manuscript is now acceptable for publication, you may indicate that here to bypass the “Comments to the Author” section, enter your conflict of interest statement in the “Confidential to Editor” section, and submit your "Accept" recommendation.

Reviewer #1: All comments have been addressed

Reviewer #2: (No Response)

2. Is the manuscript technically sound, and do the data support the conclusions?

Reviewer #1: Yes

Reviewer #2: Yes

3. Has the statistical analysis been performed appropriately and rigorously? 

Reviewer #1: I Don't Know

Reviewer #2: Yes

4. Have the authors made all data underlying the findings in their manuscript fully available?

Reviewer #1: Yes

Reviewer #2: Yes

5. Is the manuscript presented in an intelligible fashion and written in standard English?

Reviewer #1: No

Reviewer #2: Yes

6. Review Comments to the Author

Reviewer #1: I thank the authors for considering my comments and suggestions. Their response, however, was not sufficient and introduced further issues in the paper.

For example, the comments introduced regarding the added literature are often riddled with grammar errors and seem to be rushed. Specifically, lines 97-106 are simply and candidly so badly written that the level of English is unacceptable for an academic paper; it seems the authors rushed and did not properly consider and analyse the literature. Furthermore, there are still grammar mistakes here and there, but there are many grammar mistakes introduced in the newly introduced text - a professional review is advised.

Regarding the replies themselves, they are not convincing and there are comments that contradict each other.

- In Answer 1 you defend your paper by stating that it is not absolute that reinfection is even occurring, and that the objective of the model is to provide a framework for multiple strains - and not one for a single pathogen. That argument is good, but is not supported by a model that is as specific as yours (single reinfection, no exposition, etc). To argue generality, your model should be the most general. For example a SEIR model can be reduced to a SIR model and multiple reinfections from a strain can be reduced to no reinfection (just by adjusting the reinfection period to infinity). Y

I am not suggesting that you re-do your model, but that you provide an honest discussion of the limitations when comparing to other literature.

- Answer 2: Please include the discussion comparing your model to Arruda et al (2021) in the text.

The answer regarding optimal control is not convincing - observe that the control in Arruda et al and in other references for that matter, is not specific for COVID and does not include specific mitigation efforts. Rather, it is general and simply assesses the required level of infection prevention - that seems rather abstract and general to this reviewer. Are you suggesting that their model - which is arguably more general than yours in many aspects - though they do not consider the order of infection - is less realistic than yours? Again, please include a thorough and honest comparison in your paper.

Reviewer #2: I appreciate the authors' willingness to take my comments and suggestions and apply them to the revised manuscript. However, I have one final comment:

Line 406: A two-tailed t-test and a confidence interval are two different things: one deals with inference and the other deals with estimation. If you want to use the result of the two-tailed t-test, then report a p-value. If you want to use the confidence interval approach, indicate that you solved for the two-sided confidence interval of the difference between the baseline and the most aggressive strain and found that zero is not included in the confidence interval, hence you can conclude that the two are sets of data are different.

Bear in mind that a two-tailed t-test/two-sided confidence interval has an inequality for its alternative hypothesis. If your goal is to prove that it could be an upper limit, then maybe a one-sided t-test with a "greater than" alternative hypothesis would be more appropriate OR a one-sided confidence interval.

7. PLOS authors have the option to publish the peer review history of their article (what does this mean?). If published, this will include your full peer review and any attached files.

Reviewer #1: No

Reviewer #2: No

---

## [Author Response · Author response to Decision Letter 1]

2 Mar 2022

First of all, we would like to thank the reviews once again for the careful review and for making sure we produce the best academic manuscript we can. It is honestly very appreciated to get such a detailed review. 

Reviewer #1

Comment 1: “The comments introduced regarding the added literature are often riddled with grammar errors and seem to be rushed. Specifically, lines 97-106 are simply and candidly so badly written that the level of English is unacceptable for an academic paper; it seems the authors rushed and did not properly consider and analyze the literature. Furthermore, there are still grammar mistakes here and there, but there are many grammar mistakes introduced in the newly introduced text - a professional review is advised.”

Answer 1: Thank you for pointing it out. We took this comment very seriously and used professional language editing services to proofread our manuscript. Thus, we hope that this version of the manuscript is error-free. 

Comment 2: “In Answer 1 you defend your paper by stating that it is not absolute that reinfection is even occurring, and that the objective of the model is to provide a framework for multiple strains - and not one for a single pathogen. That argument is good but is not supported by a model that is as specific as yours (single reinfection, no exposition, etc). To argue generality, your model should be the most general. For example, an SEIR model can be reduced to a SIR model and multiple reinfections from a strain can be reduced to no reinfection (just by adjusting the reinfection period to infinity). I am not suggesting that you re-do your model, but that you provide an honest discussion of the limitations when comparing to other literature.”

Answer 2: You are right - we appreciate the clarification of your previous comment. Indeed, our model can be further extended by introducing expose phase (that one can reduce for specific cases by assuming E->I rate of 1). Similarly to the SIS to SIR comparison. As such, we introduce a more complete discussion on this subject in the Introduction Section. In particular, we highlight the differences between the proposed model and other multi-strain models including the strength and limitations of the proposed model. 

Comment 3: “Please include the discussion comparing your model to Arruda et al (2021) in the text. The answer regarding optimal control is not convincing - observe that the control in Arruda et al and in other references for that matter, is not specific for COVID and does not include specific mitigation efforts. Rather, it is general and simply assesses the required level of infection prevention - that seems rather abstract and general to this reviewer. Are you suggesting that their model - which is arguably more general than yours in many aspects - though they do not consider the order of infection - is less realistic than yours? Again, please include a thorough and honest comparison in your paper.”

Answer 3: The comparison between Arruda et al. (2021) and the proposed model is now provided in the Introduction Section with more details. Moreover, we extended the discussion of the usage of infection prevention in multi-strain models and clearly state that the proposed work is not handling this complexity as it would be addressed in future work based on ideas from other models such as (Reluga, 2008; Arruda et al., 2020; Edilson et al. 2021) but not limited to. 

Reviewer #2

Comment 1: “Line 406: A two-tailed t-test and a confidence interval are two different things: one deals with inference and the other deals with estimation. If you want to use the result of the two-tailed t-test, then report a p-value. If you want to use the confidence interval approach, indicate that you solved for the two-sided confidence interval of the difference between the baseline and the most aggressive strain and found that zero is not included in the confidence interval, hence you can conclude that the two are sets of data are different. Bear in mind that a two-tailed t-test/two-sided confidence interval has an inequality for its alternative hypothesis. If your goal is to prove that it could be an upper limit, then maybe a one-sided t-test with a "greater than" alternative hypothesis would be more appropriate OR a one-sided confidence interval.”

Answer 1: You are absolutely right - thank you very much for shedding some light on this issue. Following this comment, we recompute a one-sided confidence interval on the data to make sure the dynamics are indeed stat’ different and that the most aggressive strain is an upper boundary of the baseline dynamics for our data. Following that, we alter the relevant text stating it as the reviewer suggested to make sure it is an accurate description and to allow others to easily reproduce and use our results.

---

## [Decision Letter · Decision Letter 2]

7 Apr 2022

Generic Approach For Mathematical Model of Multi-Strain Pandemics

PONE-D-21-35947R2

Dear Dr. Lazebnik,

We’re pleased to inform you that your manuscript has been judged scientifically suitable for publication and will be formally accepted for publication once it meets all outstanding technical requirements.

Kind regards,

Martial L Ndeffo Mbah, Ph.D

Academic Editor

PLOS ONE

Additional Editor Comments (optional):

Reviewers' comments:

Reviewer's Responses to Questions

**Comments to the Author**

1. If the authors have adequately addressed your comments raised in a previous round of review and you feel that this manuscript is now acceptable for publication, you may indicate that here to bypass the “Comments to the Author” section, enter your conflict of interest statement in the “Confidential to Editor” section, and submit your "Accept" recommendation.

Reviewer #1: All comments have been addressed

Reviewer #2: All comments have been addressed

2. Is the manuscript technically sound, and do the data support the conclusions?

Reviewer #1: Yes

Reviewer #2: Yes

3. Has the statistical analysis been performed appropriately and rigorously? 

Reviewer #1: Yes

Reviewer #2: Yes

4. Have the authors made all data underlying the findings in their manuscript fully available?

Reviewer #1: Yes

Reviewer #2: Yes

5. Is the manuscript presented in an intelligible fashion and written in standard English?

Reviewer #1: Yes

Reviewer #2: Yes

6. Review Comments to the Author

Reviewer #1: Thanks for revising the paper and for considering my comments. All my concerns have been addressed and as far as I am concerned, the paper is ready for publication.

Reviewer #2: (No Response)

7. PLOS authors have the option to publish the peer review history of their article (what does this mean?). If published, this will include your full peer review and any attached files.

Reviewer #1: No

Reviewer #2: No

---

## [Editor Report · Acceptance letter]

20 Apr 2022

PONE-D-21-35947R2 

Generic Approach For Mathematical Model of Multi-Strain Pandemics 

Dear Dr. Lazebnik:

I'm pleased to inform you that your manuscript has been deemed suitable for publication in PLOS ONE. Congratulations! Your manuscript is now with our production department. 

Kind regards, 

on behalf of

Dr. Martial L Ndeffo Mbah 

Academic Editor

PLOS ONE